# High-Resolution Color Transparent Display Using Superimposed Quantum Dots

**DOI:** 10.3390/nano12091423

**Published:** 2022-04-21

**Authors:** Mahboubeh Dolatyari, Farid Alidoust, Armin Zarghami, Ali Rostami, Peyman Mirtaheri, Hamit Mirtagioglu

**Affiliations:** 1SP-EPT Lab., ASEPE Company, Industrial Park of Advanced Technologies, Tabriz 5364196795, Iran; mdolatya@uni-koeln.de (M.D.); zarghamiarmin@gmail.com (A.Z.); 2OIC Research Group, Faculty of Electrical and Computer Engineering, University of Tabriz, Tabriz 5166614761, Iran; farid.alidoost@gmail.com; 3Department of Mechanical, Electronics and Chemical Engineering, OsloMet—Oslo Metropolitan University, 0167 Oslo, Norway; 4Department of Statistics, Faculty of Science and Literature, University of Bitlis Eren, Bitlis 13100, Turkey; hmirtagioglu@beu.edu.tr

**Keywords:** transparent display, quantum dots, superimposed, nanomaterials

## Abstract

In this paper, a high-resolution full-color transparent monitor is designed and fabricated using the synthesized quantum dots for the first time. For this purpose, about 100 compounds that had the potential to emit blue, green, and red lights were selected, and simulation was performed using the discrete dipole approximation (DDA) method, in which the shell layer was selected to be SiO_2_ or TiO_2_ in the first step. Among the simulated compounds with SiO_2_ or TiO_2_ shells, Se/SiO_2_ and BTiO_3_/SiO_2_ were selected as blue light emitters with high intensity and narrow bandwidth. Accordingly, CdSe/SiO_2_ nanoparticles were selected as green light emitters and Au/TiO_2_ for the red light. As the surface of the nanoparticles in their optical properties is important, reactivation of the nanoparticles’ surface is required to reach the high-intensity peak and resolution. To this end, in the second step, the surface of Se and CdSe nanoparticles reacted with ethanolamine, which can make a strong bond with cadmium atoms. The band structure and optical properties were obtained by the *density functional theory* (DFT) method. The Se/Ethanolamine and CdSe/Ethanolamine were experimentally synthesized to evaluate the theoretical results, and their optical properties were measured. To fabricate a transparent monitor, Se/Ethanolamine, CdSe/SiO_2_, and Au/TiO_2_ nanoparticles were dispersed in polyvinyl alcohol (PVA) solved in water and deposited on the glass by the doctor blading technique. Finally, high-resolution videos and images were displayed on the fabricated monitor.

## 1. Introduction

Recent advances in the semiconductor industry and the demands for more screen-based entertainment, intellectual methods of interaction, and groundbreaking visualization techniques have led to a large interest in nontraditional display designs. Transparent displays would have useful applications in aircraft cockpits or car windshields, surgical procedures, entertainment industries, smart advertisements, and others [1]. There are a few kinds of displays; the simplest is a head-up display, and its limitation is a narrow viewing angle [1,2], which can be improved using diffusive screens that have no selectivity in wavelength. To achieve strong scattering, the screen’s transparency would become weak. Furthermore, limitations in transparency can be improved by using frequency conversion displays that comprise the materials that convert both ultraviolet emission and infrared to visible light [1,3,4,5,6,7]. Moreover, there are also electronic flat panels that are based on LEDs with transparent electrodes that do not rely on projection; therefore, achieving high-frequency operations is difficult. [8,9]. The major limitation is making the large size of displays. In recent years, new types of transparent displays based on highly efficient nanoparticles have wide viewing angles and scalability to large sizes [1,10]. Finding suitable material with a high scattering cross-section is essential for increasing efficiency and reducing production costs. For this purpose, core-shell QD structures were used to develop transparent displays. For instance, Ag/SiO_2_ QDs, introduced by C. W. Hsu can cover the full visible spectrum by changing the size of QDs [1]. Practically, it is observed that by increasing the size of the nanoparticles, the intensity of scattered light decreases, and thus the resolution of the images will become low. Therefore, applying one type of nanoparticle with different sizes cannot make high-resolution images, and finding a fluorescent material with high intensity is one of the major challenges. Improving the efficiency of blue light with Si/SiO_2_ QDs, which shows high-resolution images in blue wavelengths, has been reported previously by our group. The important advantages of this work are the low cost, high resolution, and easy mass production [10]. To make the color transparent display, applying superimposed ideas can create the possibility of making a crystalline lattice that can scatter different colors from one point to another. Designing such a monitor is not possible with classic methods. However, with the synthesis of the QDs (with high intensity of light scattering in blue, green, and red wavelengths), mixing the selected materials with specific density, and then spin coating on the surface of the glass, we can reach a crystalline layer with an ability to scatter colors. To find suitable nanoparticles, we analyzed more than one hundred different materials embedded in a core-shell structure by sweeping diameter and any combination of fractions in the core-shell by the DDA theoretical method. Since the calculated results are too numerous to be discussed thoroughly in this paper, we selected some simulated nanomaterials to describe their properties. Among all the investigated materials, in the first step, Se/SiO_2_ and BaTiO_3_/SiO_2_ nanoparticles are selected as blue, CdSe/SiO_2_ quantum dot as green, and Au/TiO_2_ nanoparticles as red QDs for synthesis. In the second step, the surface of nanoparticles is also coated with ethanolamine to reveal the role of the surface of the nanoparticles and the electronic trap levels in their structures for increasing the quantum yield and intensity of the light scattering.

## 2. Methodology

### 2.1. Mathematical Modeling

The DDA method was developed to investigate the interaction of electromagnetic waves (EM) with objects in any environment. The efficiency of QDs is defined as cross-section quantity, which is the net rate of electromagnetic energy scattering (Equation (1)) from the surface of a hypothetical sphere of radius *r* ≥ R centered on the particle origin divided by incident irradiance and area of the particle (Equation (2)) [11,12,13].
(1)$$W=-\int_{A}\left(E\times H\right)\cdot \hat{n}dA$$
(2)$$\sigma_{Scat}=\frac{\omega_{Sat}}{I_{i}\times \pi r^{2}}$$

The scattering cross-section depends upon the wavelength of the incident light, permittivity, shape, and size of the QD nanoparticle. The total amount of scattering in a sparse medium is proportional to the product of the scattering cross-section and the number of particles present in the medium. Regarding the DDA method, if there is an incident field *E_inc,j_* on each dipole, there will be filed contributions from other re-radiating dipoles. QDs are believed to radiate instantly because of the proximity of dipoles they induce. By calculating the incident E-fields on each dipole (*E_inc,j_*) plus its interaction with other N-1 dipoles, we can calculate the amplitude (*E_j_*) of the time-harmonic E-field at each dipole position (*r_j_*).

Then, the system equation can be constructed as Equation (3) [14].
(3)$$E_{j}=E_{inc,j}-\sum_{k\neq j}A_{jk}P_{k}$$
where *A_jk_* is the tensor that illustrates the interaction between a receiving dipole *r_j_* and the radiating dipole at *r_k_*. The off-diagonal block 3 × 3 tensors in the interaction matrix are of the form Equation (4) [15].
(4)$$A_{jk}=\frac{e^{ik_{jk}.r}}{r_{jk}}\times \left[k^{2}\left(r_{jk}r_{jk}-I_{3}\right)+\frac{ik.r_{jk}-1}{r_{jk}{}^{2}}\left(3r_{jk}r_{jk}-I_{3}\right)\right]$$

Equation (3) can be simplified to Equation (5), which leaves us with solving 3N unknown dipole moments *Pj* in the precisely determined system of 3N linear equations [15].
(5)$$\sum_{k=1}^{N}A_{jk}P_{j}=E_{inc,j}$$

Once Equation (5) is solved, extinction and absorption cross-sections, *C_ext_* and *C_abs_* can be calculated using Equation (6), where ζ is the imaginary part, and *E*_0_ is the amplitude of the incident wave. Then, the scattering cross-section can easily be calculated by differentiating extinction and absorption cross-sections [15].
(6)$$C_{ext}=\frac{4\pi k}{{\left|E_{0}\right|}^{2}}\overset{N}{\underset{j=0}{\sum }}\zeta \left(E_{inc,j}\times P_{j}\right)$$
(7)$$C_{abs}=\frac{4\pi k}{{\left|E_{0}\right|}^{2}}\sum_{j=0}^{N}\left\{\zeta [\left(P_{j}\left(\alpha_{j}{}^{-1}\right)\times P_{j}\right]-\frac{2}{3}k^{3}{\left|P_{j}\right|}^{2}\right\}$$
(8)$$C_{Scat}=C_{ext}-C_{abs}$$

In our optimization of the core-shell structure, we utilized different materials with an appropriate fixed shell as the first step. The plasmonic effect will occur as our materials are either metals or semiconductors; therefore, it is better to use a dielectric shell that must be transparent to the visible spectrum. We investigated different materials in the nanometer regime and repeated analysis and scattering calculations to find the best material to solve this problem. We came up with silicate (SiO_2_) based on the high Q-factor of the nanoparticles containing silicate. In a second approach, to achieve a core-shell structure appropriate for visible light, we can utilize a material with a high scattering potential as a core and place different materials in the shell. The core material of TiO_2_ in this approach is selected to be a fixed material in nanoparticles optimization. The lattice constant of the core and shell materials must match in both approaches. Furthermore, the materials chosen for fabrication must be able to undergo the solution process method. Needless to say, the materials chosen should be affordable, accessible, and environmentally friendly. On the first approach, one hundred forty various materials were tasted as the core, and on the second approach, a few different materials were tested as the shell. In our core-shell QD, the diameter of that in each test is shifted in the range of 5 to 155 nm with a sequence of 15 nm. In addition, the ratio of the core to the shell was changed in every test by the following conditions (0.1, 0.3, 0.5, 0.7, and 0.9). A total number of 10,800 analyses with the beforementioned structure for both proposed approaches was performed. Considering that the foundation of applied analyses was performed through the DDA method, to guarantee the results obtained from the DDA method, some of the QDs were chosen randomly for evaluation in the FEM and the FDTD methods. The same process was used for some samples that drew our attention. By comparing the results, we understand that resonance frequencies are the same in all methods and the maximum error was 5 percent in the Q domain, which is acceptable.

### 2.2. Fabrication Process Methods

#### 2.2.1. Materials

All chemicals (of analytical grade) were purchased from Merck and used without further purification. 

#### 2.2.2. Characterization

The morphology of products was studied via a Tscan model MIRA3 field emission scanning electron microscope with the accelerating voltage of 10 kV. TEM images were obtained on a Zeiss-EM10C-80KV transmission electron microscope with an accelerating voltage of 80 kV. UV–Vis absorption spectra were recorded employing a PG instruments Ltd. T70 UV/Vis spectrophotometer. Photoluminescence measurements were carried out by a Perkin Elmer LS55 luminescence spectrophotometer.

The crystalline quality of the as-prepared ZnO nanoplates was analyzed by X-ray diffraction on a Siemens D5000 using a Cu-k_α_ line of wavelength λ = 1.541 Å at the scanning rate of 1°/min. 

#### 2.2.3. Synthesis Method

*Synthesis of Se/PVP nanomaterials:* First, 1.405 g selenium dioxide and 3 g PVP were dissolved in 250 mL distilled water, and then ascorbic acid was solved in 250 mL distilled water and mixed with the first solution at room temperature. Afterward, the obtained selenium particles were centrifuged three times with distilled water, ethanol, and acetone. 

*Synthesis of Se/SiO_2_ nanomaterials:* The obtained Se/PVP particles were dispersed in ethanol, and 0.025 g tetraethyl orthosilicate and 1 cc hydrochloric acid (0.1 M) were injected into the flask and stirred for 10 min. Then, 0.01 g ammonia (1 M) was injected into the flask and stirred for 15 min. The obtained particles were dispersed in ethanol using an ultrasonic bath. The obtained particles were washed several times with ethanol, water, and acetone to eliminate all impurities.

*Synthesis of Se/Ethanolamine nanomaterials:* The obtained Se/PVP particles were dispersed in ethanol, and 0.1 g of ethanolamine was added and stirred for more 24 min. The obtained particles were washed several times with ethanol, water, and acetone to eliminate all impurities. 

*Synthesis of BaTiO_3_/SiO_2_ nanomaterials:* One mmol barium chloride was dissolved in ethanol at 65 °C, and the solution was cooled down to room temperature. Then, 50 cc 2-propanol and 10 cc mono ethylene glycol were introduced to the solution. In the next step, adding 1 mmol of titanium isopropoxide causes partial hydrolysis. The mixture was subjected to low temperature (2–3 °C) and simultaneously, deionized water was introduced. During all steps of sol preparation, a magnetic stirrer agitated the mixture. The obtained particles dispersed in ethanol and 0.025 g tetraethyl orthosilicate and 1 cc hydrochloric acid (0.1 M) were injected into the flask and stirred for 10 min. Then, 0.01 g ammonia (1 M) was injected into the flask and stirred for 15 min. The obtained particles were washed several times with ethanol, water, and acetone to eliminate all impurities. The obtained particles were dispersed in ethanol using an ultrasonic bath. 

*Synthesis of CdSe/SiO_2_ nanomaterials:* First, 1.63 g Cd(NO_3_)_2_ and 1 g oleic acid were mixed and heated to dissolve the Cd(NO_3_)_2_ powder under a nitrogen atmosphere at 150 °C. Then, 0.395 g selenium was solved in 3g trioctylphosphin (TOP) [16] and injected into the flask containing cadmium nitrate at the temperature of 70 °C. After 1 h, 0.025 g tetraethyl orthosilicate was injected into the flask and stirred for 10 min. Then, 0.01 g ethanolamine was injected into the flask and stirred for 15 min. The obtained particles were washed several times with ethanol, water, and acetone to eliminate all impurities. Afterward, they were dispersed in ethanol using an ultrasonic bath. 

*Synthesis of CdS**e**/Ethanolamine nanomaterials*: First, 1.63 g Cd(NO_3_)_2_ and 1 g oleic acid were mixed and heated to dissolve the Cd(NO_3_)_2_ powder under a nitrogen atmosphere at 150 °C. About 0.395 g selenium was solved in 3 g TOP and injected into a flask containing cadmium nitrate at the temperature of 70 °C. Afterward, 0.01 g ethanolamine was injected into the flask and stirred for 15 min. The obtained particles were then washed with n-hexane, ethanol, water, and acetone several times to eliminate all impurities. The obtained particles were dispersed in ethanol, and 0.1 g ethanolamine was added to it and stirred for more than 24 min. After that, the particles were washed with n-hexane, ethanol, water, and acetone multiple times to eliminate all impurities. 

*Synthesis of Au/TiO_2_ nanomaterials*: In one flask, 0.052 g HAuCl_4_ was dissolved in 100 mL deionized water, and in a second flask, 0.5 g sodium citrate was dissolved in 100 mL deionized water and heated up to 100 °C. The gold solution was injected into the second flask and stirred for 1 h. The solution was cooled down and stored at room temperature [17]. Finally, a mixture of 0.001 g titanium isopropoxide was dissolved in 5 cc mono ethylene glycol, and 15 cc isopropyl alcohol was added by dripping on the gold solution and stirred for 15 min. The obtained colloid was used as red fluorescent material without any special treatment.

*Fabrication method*: The obtained nanomaterials were dispersed in ethanol or water and added to polyvinyl alcohol (PVA) dissolved in water. The attained viscose liquid was put on glass and dried for 12 h according to the doctor blading method [10]. 

### 2.3. DFT Calculations

The electronic band structure and the optical properties of Se/SiO_2_, Se/Ethanolamine, CdSe/SiO_2,_ CdSe/Ethanolamine, and Au/TiO_2_ quantum dots were calculated using DFT calculations. The calculations were performed with the CASTEP code [18] and optimized using the Broyden–Fletcher–Goldfarb–Shanno (BFGS) geometry optimization method [18]. Generalized gradient approximation (GGA) and the non-local gradient-corrected exchange-correlation functional as parameterized by Perdew–Burke–Ernzerhof (PBE), which uses a plane wave basis set for the valence electrons and norm-conserving pseudopotential (NCP) [19,20] for the core electrons, were used in the calculation. The number of plane waves included in the basis was determined from the cut-off energy (*E*c) of 500.0 eV. The summation over the Brillouin zone was carried out with *k*-point sampling using a Monkhorst-Pack grid with parameters of 2 × 2 × 2. Geometry optimization under applied hydrostatic pressure can be used to determine the modulus of a material (B) and its pressure derivative, B’ = dB/dP. 

## 3. Results and Discussion

For more than 100 nanomaterials with different radiuses, the absorption and scattering calculations have been performed using the DDA method for achieving high intense blue, green, and red scattering. Some cases were investigated, and the appropriate nanomaterials were selected to fabricate a transparent display. The results are illustrated in Figure 1.

Among the calculated materials, the selected nanoparticles are listed in Table 1; some of them have potentially applicable for synthesis and use in the fabrication of transparent displays. The absorption and scattering of the listed materials are shown in Figure 2. 

Between all these selected materials, we synthesized Se/SiO_2,_ Se/Ethanolamine, and BaTiO_3_/SiO_2_ as blue, CdSe/SiO_2_ as green, and Au/SiO_2_ as red scattered materials, and their characterization and physical investigations are described in this part. 

### 3.1. Studies on Blue Emission

The optical properties of selenium sol have been investigated by several groups [21,22,23,24]. As Figure 3 shows, emission spectrum is narrow with high intensity. As the figure shows, the intensity of the emission spectrum is eight times more than the absorption spectrum intensity, and the standard deviation of the emission band is about 50 nm. These nanoparticles can be synthesized using the reduction of selenium oxide. In this paper, we prefer Se/SiO_2_ as blue fluorescent for introducing very interesting blue fluorescent nanomaterial that makes high-resolution images. The materials and methods section describes the synthesis method for Se/SiO_2_ nanocolloids.

Barium titanate (BaTiO_3_) has a perovskite structure with a general formula of ABO_3_, and it has been extensively studied due to its ferroelectric properties, piezoelectricity, pyroelectricity, and high dielectric constant [25]. The optical properties of BaTiO_3_ are also interesting, and it has been studied by several groups [26,27,28,29,30,31]. In our calculations, barium titanate shows satisfactory emission in a blue range (Figure 2); however, its intensity is lower than Se/SiO_2_. The simulated results indicate that the ratio of emission to absorption is more than six at 400 nm. The peak is relatively narrow and its width (standard deviation) is 30 nm. Thus, it would be suitable for applying as a blue emitter in a transparent display. Here, we could synthesize colloidal BaTiO_3_ as a material for making blue emissions, which were produced by using BaCl_2_ and TiCl_4_ in alkaline media. 

The intensity of the scattering peak for BaTiO_3_ /SiO_2_ is 6.8 and the absorption peak has an intensity of 0.024. It means the intensity of the emission band is 283.3 times more than absorption (see Figure 4). In addition, the emission band is narrow and these properties make it suitable for display applications. 

For blue emission, Se/SiO_2_ and BaTiO_3_/SiO_2_ were synthesized and characterized. The particles are spheres with a size of 60 nm. Figure 5 shows the TEM image of synthesized Se/SiO_2_ nanoparticles.

The XRD diffraction pattern indicates the hexagonal structure of the synthesized Se nanoparticles (Figure 6). The peaks are indexed to (100), (101), (110), (102), (111), and (201) lattice planes of hexagonal selenium. The pattern is in good agreement with the characteristic peaks from The International Centre for Diffraction Data (ICDD) (PDF 65-1876).

Narrowband emission facilitated the production of high-resolution images. As Figure 7 shows, Se/SiO_2_ has two absorption bands at 270 nm and 390 nm; however, we have an intense emission band at 400–450 nm with CIE coordinates of (0.17, 0.05). The density of states obtained by the DFT calculations indicates (Figure 8) the absorption band at 400 nm, and it is related to the electronic transitions between s and p orbitals of Se atoms. As the partial density of states shows, the atoms related to SiO_2_ (Si and O atoms) have minimum effect on the electronic transitions of the structure. 

Figure 9 shows the XRD pattern of synthesized BaTiO_3_/SiO_2_. The pattern indicates cubic BaTiO_3_ with a space group of pm3m according to ICDD-PDF 01-084-9618.

As Figure 10 shows, the intensity of the scattering spectrum for BaTiO_3_/SiO_2_ is two times higher than that of the absorption spectrum; therefore, it is applicable as a blue emitter for a transparent display. However, Se/SiO_2_ is more suitable for this purpose because the intensity of luminescence spectroscopy is higher with high brightness. It has a narrower band in blue wavelengths than BaTiO_3_/SiO_2_, which can be related to the band structure of the materials. 

Luminescence in the crystal lattice occurs with a few mechanisms, some of which are presented in Figure 11. The emitted light can be controlled by engineering the trap levels and impurities for making defect levels.

For nanomaterials, controlling the traps has an essential role in the intensity of emitted light. Therefore, to increase the luminescent intensity, the surface of Se nanoparticles, modified by SiO_2_, is exchanged with different ligands such as ethanolamine. Based on the results from Shaobo Ji et al., the binding of the compounds with N…H bonds and Se makes covalent solid bonds [32]. Our calculations confirm this result, and as the band structure of the Se nanoparticle with ethanolamine ligand shows, some levels have been moved from the conduction band of Se without ligand. This means that the interaction of surface Se atoms with ethanolamine makes stable bonds, and the splitting between bonding and antibonding levels is significant at the end. A greater band structure with ethanolamine can be achieved. The band structure can eliminate some trap levels created by dangling bands (see Figure 12). As Figure 13 shows, the absorption intensity of Se modified by ethanolamine ligand is 2.3 times more than Se/SiO_2_. If our emission light is according to the mechanism described in Figure 11a, eliminating trap levels from the band structure can improve the intensity of emission (See Figure 14). The results were confirmed by the scattering spectra of synthesized Se/SiO_2_ and Se/Ethanolamine nanoparticles. Figure 14 shows the emission spectra of synthesized Se/SiO_2_ and Se/Ethanolamine QDs. 

### 3.2. Studies on Green Scattering

The optical properties of CdSe/SiO_2_ were investigated by different scientists [33,34,35,36,37,38]. The theoretically obtained scattering spectrum shows an intensity of 6.2 in a normalized condition for CdSe/SiO_2_. Moreover, the intensity of the absorption peak is low (about 1.5), and it can emit a brilliant green color with absorbing low intensity of green color. Based on these results, we selected this type of quantum dot for green emission. Figure 15 shows the absorption and scattering spectra of the CdSe/SiO_2_ nanoparticles. 

The TEM image of the synthesized CdSe/SiO_2_ shows that the average size of the nanoparticles is about 15 nm (Figure 16). 

Figure 17 shows the XRD pattern of the synthesized CdSe/SiO_2_ nanoparticles. The observed peaks at 2θ, 25.2, 26.4, 28.3, 36.7, 43.7, 48, 51.8 can be assigned as (100), (002), (101), (110), (103), (200), (112), and (211) crystalline planes, respectively, which are according to the hexagonal CdSe structure reported in ICDD PDF 02-0330 (Appl. Sci. 2016, 6, 278). 

The absorption peaks in CdSe/SiO_2_ are relatively broad at 300–350 nm and 450–540 nm (Figure 18). The scattering spectrum is relatively narrow at 520 nm with CIE coordinates of (0.08, 0.83), and its profile’s standard deviation is 15 nm, resulting in a high-resolution scattering. The calculated density of states indicates electronic transitions are between the p orbitals of Se atoms and d orbitals of Cd atoms for the CdSe crystals. Covering the surface of CdSe by SiO_2_, it can be strengthened by the p orbitals of O atoms (See Figure 19).

Moving trap levels to higher energies caused a shift in the wavelength of the fluorescence. By exchanging SiO_2_ with ethanolamine as a shell, the peak observed at 550 nm shifted to shorter wavelengths (absorption band: 350 nm emission band 500 nm) (Figure 20A,B); however, the absorption peak intensity at 250 nm increased. This means that the scattering of green light can be generated by the mechanism shown in Figure 19B. In this study, we needed green light, and the amine ligand was not helpful for our application. 

### 3.3. Studies on Red Scattering Light

Gold nanoparticles have interesting optical properties, such as surface plasmon resonance (SPR) [39,40,41], sensing [42,43,44], and photonics [45], which makes for fascinating practical applications in various fields [46]. The most challenging part was red scattering because most materials had no narrowband scattering in this range and had a scattering of green light besides the red color. Although its peak intensity is not so high (about 3.8), and it is two times higher than its absorption intensity (Figure 21), Au/TiO_2_ was better than others and thus, selected in this study. Due to the higher scattering response induced by the SPR effect and narrower bandwidth, gold nanoparticles with titania shells were applied to fabricate the transparent display described in this paper. 

The TEM image of the Au/TiO_2_ nanoparticles is shown in Figure 22, which shows that the size of the nanoparticles is about 30 nm. 

The XRD pattern of the synthesized Au/TiO_2_ nanoparticles is shown in Figure 23. The peaks at 2θ = 38.2, 44.1, and 64.4 correspond to standard Bragg reflections (111), (200), and (220) of face-centered cubic (fcc) lattice related to Au lattice crystal (ICDD PDF-99-0056).

As Figure 24 shows, Au/TiO_2_ has a broadband at 540 nm; however, the luminescent band is narrow (standard deviation is 10 nm), and the peak is located at 600 nm with CIE coordinates of (0.63, 0.37). Electronic transition for this band can be allowed between the d orbitals of Au atoms and the p and s orbitals of Au atoms. However, the p orbitals of O atoms can help increase the intensity of this transition (Figure 25).

### 3.4. Fabricated Transparent Monitor

Figure 26 shows the simulation structure’s overall geometry. The simulation structure is triggered with a white light source in the range of 400 to 700 nm. There is a pattern under the triggering source consisting of optical filters and a perfect-electric-conductor (PEC). Figure 27 shows the extinction coefficient of colored band-pass filters from HOYA Co. This structure mimics the behavior of a projector, and in the last image, we exhibit quantum dots that aggregate together on a silicate substrate.

This simulation uses air, SiO_2_, AlSb, GaP, HgS, and PVP. Materials data, such as dielectric functions (ϵ) and refractive index (n), can be found in the COMSOL library and the refractiveindex.info website. The silicate substrate layer thickness is 2 μm.

The wave equation responsible for this simulation is found in the physics module of electromagnetic waves, frequency domain (EWFD), as follows.
(9)$$\nabla \times (\nabla \times E)-k_{0}^{2}\epsilon_{r}E=0$$
(10)$$E(x,y,z)=\tilde{E}(x,y)e^{-ikz^{z}}$$

With COMSOL, our transparent display model could be excited with varying wavelengths across the visible spectrum to run simulations with varying excitation wavelengths for the same geometry. Three primary wavelengths were selected, 440 nm, 520 nm, and 630 nm, to demonstrate the ability of the transparent display to scatter multiple colors when a visible range of incident light is applied.

A slice plot in Figure 28 illustrates the scattered electric field amplitude over the nanoparticles. A plot of the normalized electric field over the quantum dots is illustrated in an x-y plane and depicts the magnitude and direction of the scattered field over those QDs. The simulation shows that the proposed transparent display has a high resolution because the patterns of light projected on the display are well captured and re-emitted with high efficiency in the opposite direction of the incident light. Figure 28 shows that the spatial accuracy in both the X and Y directions can be defined as at least one micrometer in such a way that the scattered light does not interact with the neighboring quantum dots.

An illustration of how our proposed full-color transparent display will perform is exemplified in Figure 29. The synthesized QDs were dispersed in PVA and deposited on the surface of a glass substrate using the doctor blade technique to form a thin layer of QDs on the glass. As it is noticeable, a response from the fabricated transparent display presents a sharp and high-resolution image, and there is no blending of colors in the adjacent regions.

## 4. Conclusions

Absorption and scattering spectra for more than 100 compounds that had the potential to emit blue, green, and red lights were simulated using the discrete dipole approximation (DDA) method. Among the simulated compounds, Se/SiO_2_, Se/Ethanolamine, and BaTiO_3_/SiO_2_ (as blue), CdSe/SiO_2_ (as green), and Au/TiO_2_ (as red) were selected and synthesized by the solution-processed method. The synthesized particles were characterized and used for fabricating a transparent display. The synthesized colloidal nanoparticles were dispersed in PVA and deposited on the glass substrate. The fabricated display was tested for monitoring images and movies. The resolution of the displayed images and movies indicates a high and bright qualification of the fabricated monitor.

## Figures and Tables

**Figure 1 nanomaterials-12-01423-f001:**
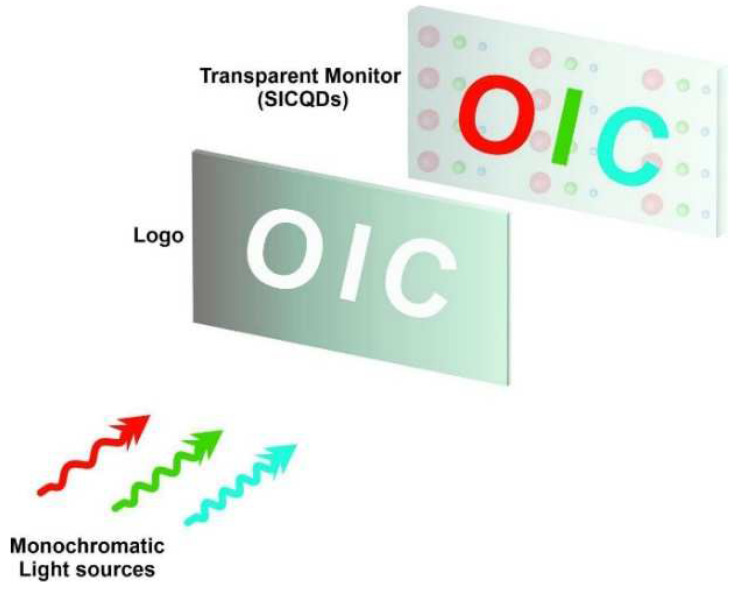
Image formation on display using a logo and monochromatic light sources.

**Figure 2 nanomaterials-12-01423-f002:**
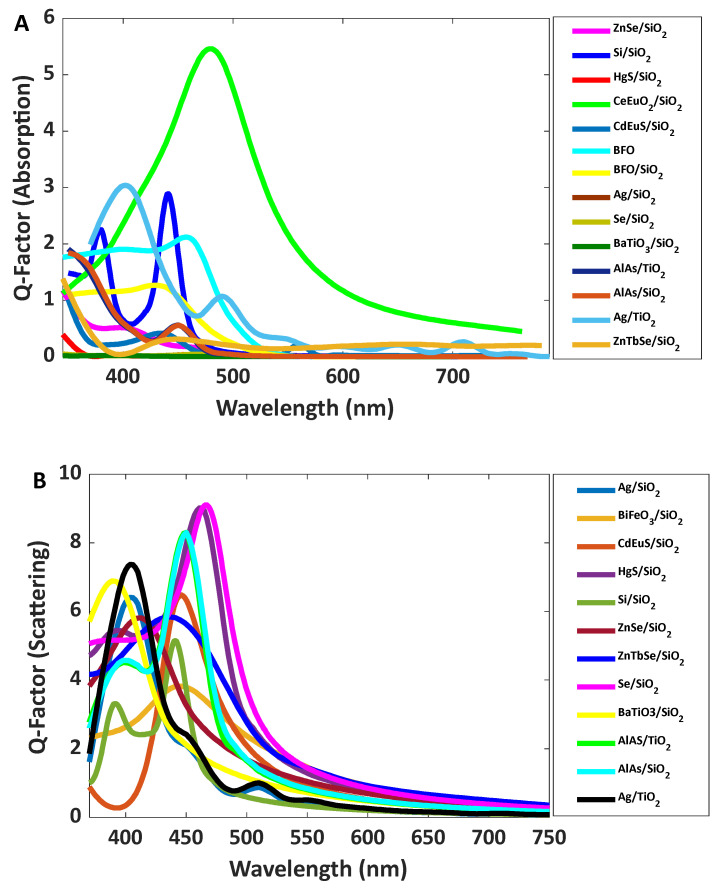
(**A**) Absorption and (**B**) scattering spectra of simulated nanoparticles for blue color; (**C**) absorption and (**D**) scattering spectra of simulated nanoparticles for green color; and (**E**) absorption and (**F**) scattering spectra of simulated nanoparticles for red color.

**Figure 3 nanomaterials-12-01423-f003:**
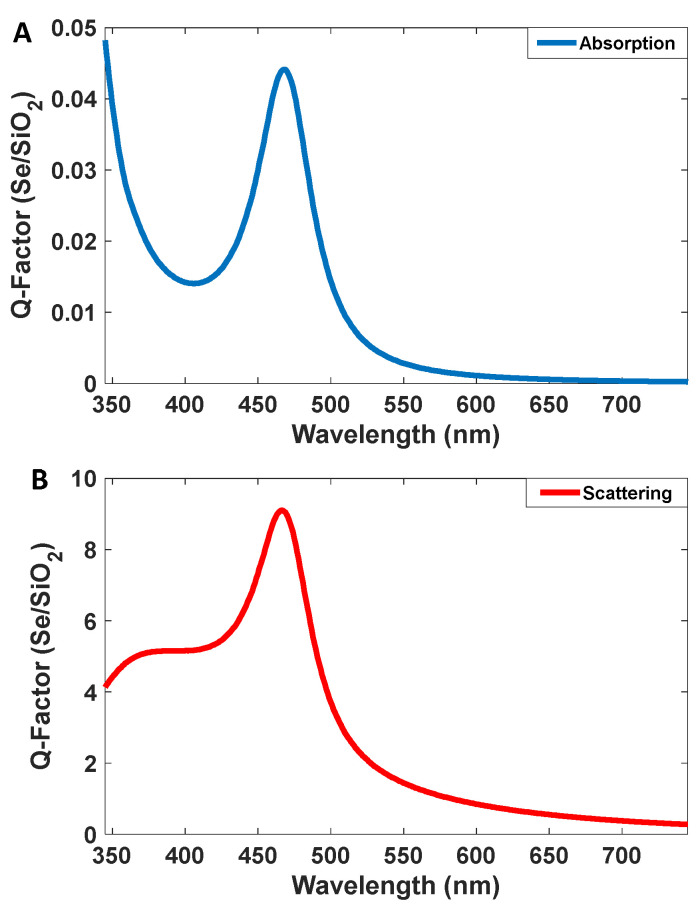
(**A**) Absorption and (**B**) emission spectra of the Se/SiO_2_ nanoparticles.

**Figure 4 nanomaterials-12-01423-f004:**
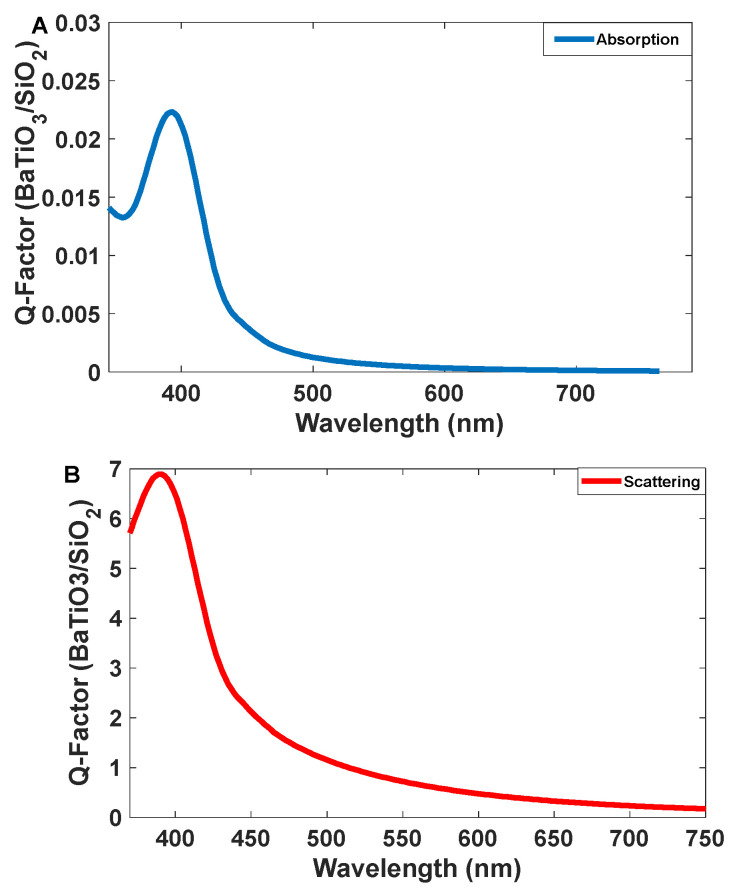
(**A**) Absorption and (**B**) scattering spectra of the BaTiO_3_/SiO_2_ nanoparticles.

**Figure 5 nanomaterials-12-01423-f005:**
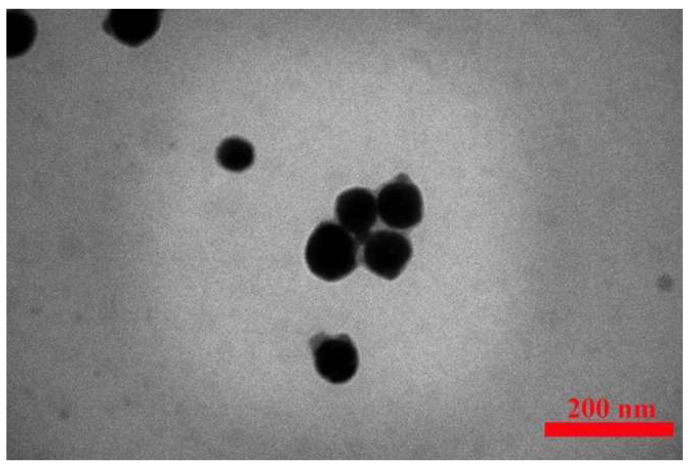
TEM image of the synthesized Se/SiO_2_ nanoparticles.

**Figure 6 nanomaterials-12-01423-f006:**
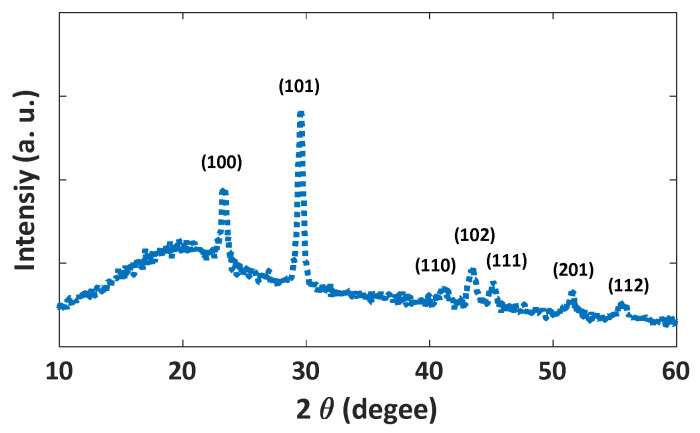
The XRD diffraction pattern of the synthesized Se/SiO_2_ nanoparticles.

**Figure 7 nanomaterials-12-01423-f007:**
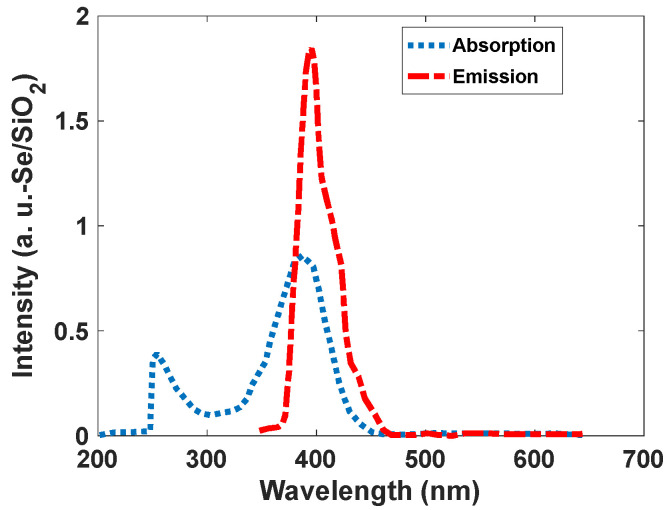
Absorption and emission spectra of the synthesized Se/SiO_2_ nanoparticles.

**Figure 8 nanomaterials-12-01423-f008:**
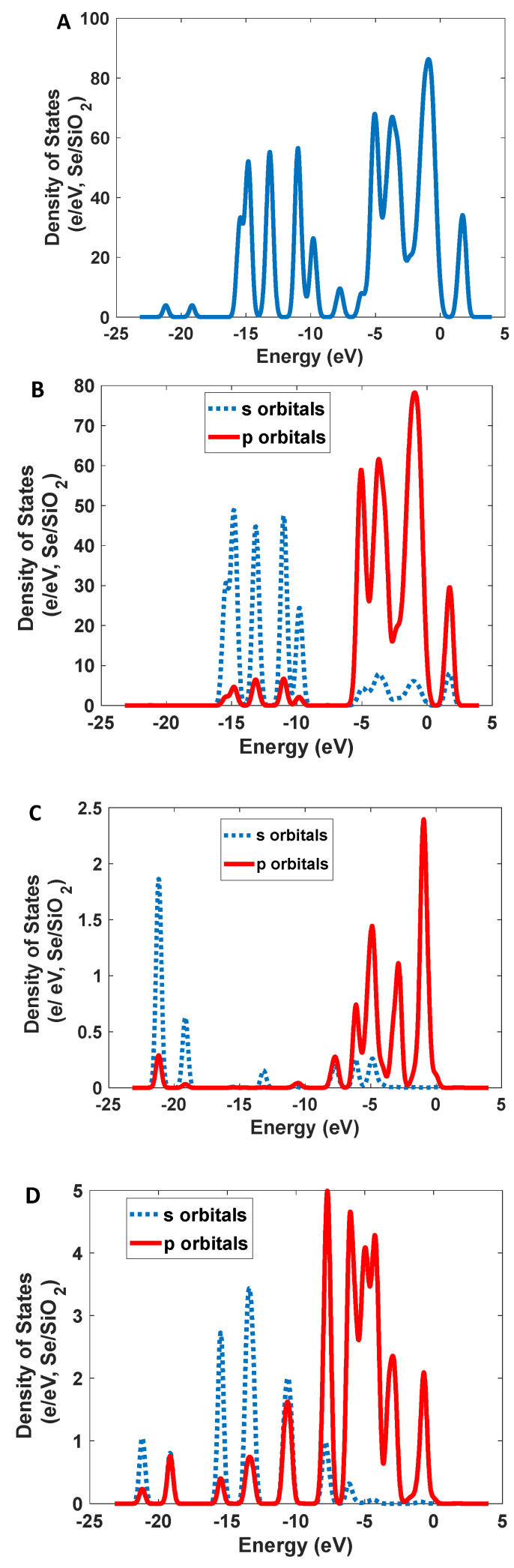
The calculated density of states for Se/SiO_2_ nanoparticles: (**A**) total; (**B**) Se atoms; (**C**) O atoms; and (**D**) Si atoms.

**Figure 9 nanomaterials-12-01423-f009:**
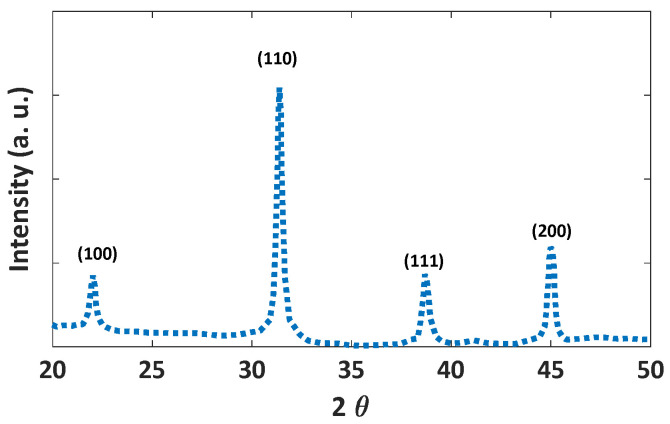
The XRD diffraction pattern of the synthesized BaTiO_3_/SiO_2_ nanoparticles.

**Figure 10 nanomaterials-12-01423-f010:**
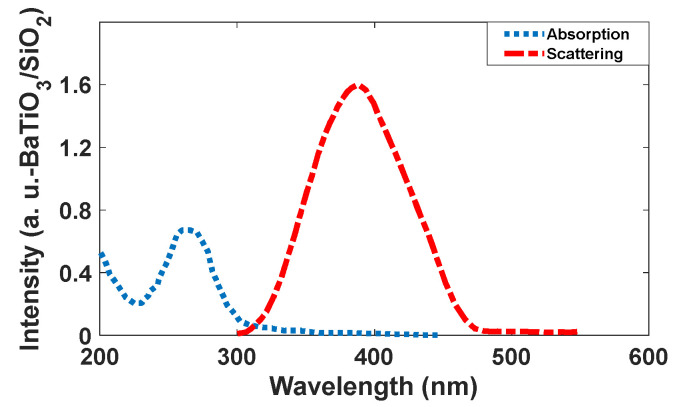
Absorption and scattering spectra of the synthesized BaTiO_3_/SiO_2_ nanoparticles.

**Figure 11 nanomaterials-12-01423-f011:**
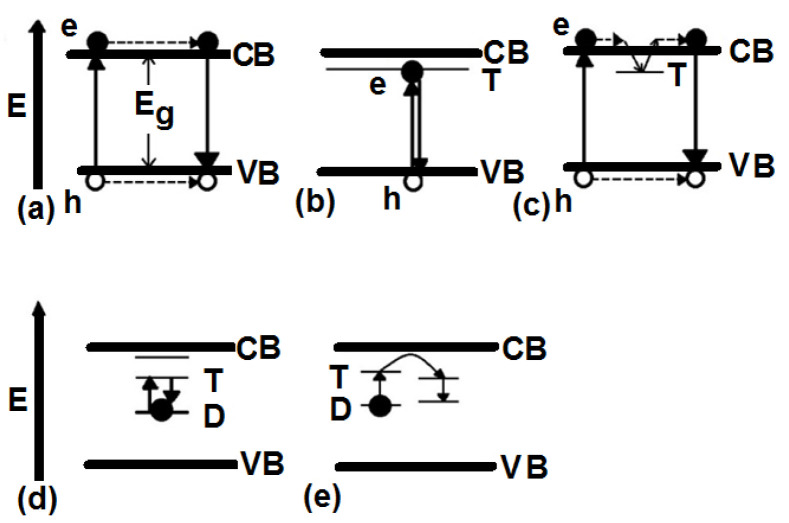
The luminescent mechanisms for different crystal lattices, (**a**) the electron in the valence band absorbs energy, transfers to the conduction band, and creates a hole in the valence band. Subsequently, the excited electron recombines with created hole by emitting a photon, (**b**) If there are trap states between the valence band and conduction band, the electron can transfer to trap levels, and then, by recombination with the hole (in the valence band) energy is emitted, (**c**) Surface defects can introduce metastable or trap levels in the energy gap. An electron raised to the conduction band may be trapped in the defect level T, instead of recombining immediately with a hole in the valence band, (**d**) Maybe there are trap levels in the energy gap and an electron, by absorbing energy, jumps from the ground to that excited level of the traps and subsequently decays by emitting energy, (**e**) The excited electron in the defect level (D as donor) can be transferred to a neighboring impurity (A, the acceptor) and makes radiative or nonradiative energy.

**Figure 12 nanomaterials-12-01423-f012:**
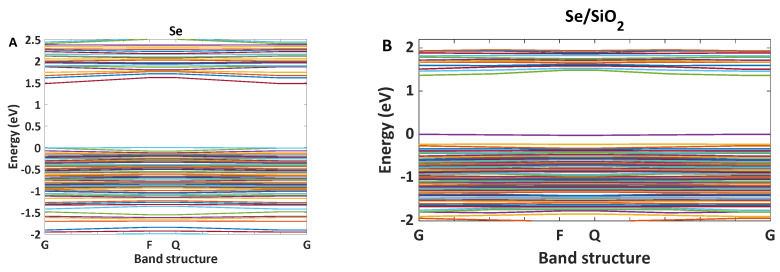
Calculated bandstructure for Se, Se/SiO_2_, and Se/Ethanolamine nanoparticles, (**A**) Se, (**B**) Se/SiO_2_, and (**C**) Se/Ethanolamine nanoparticles. Comparison between Se and Se/SiO_2_ indicates some distances between energy levels in the conduction band and valence band however, the bandgap has not changed. In the case of Se/ Ethanolamine bandgap of the Se has increased.

**Figure 13 nanomaterials-12-01423-f013:**
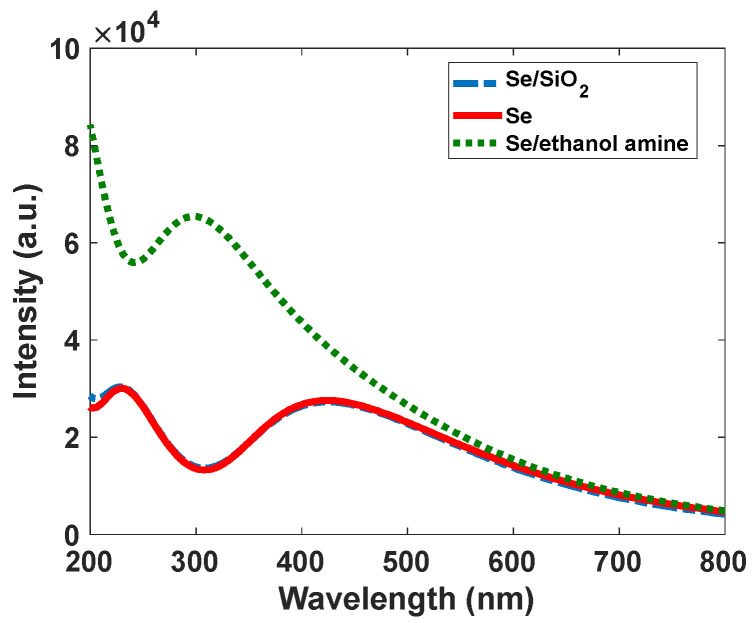
Calculated absorption spectra of Se, Se/SiO_2_, and Se/Ethanolamine nanoparticles.

**Figure 14 nanomaterials-12-01423-f014:**
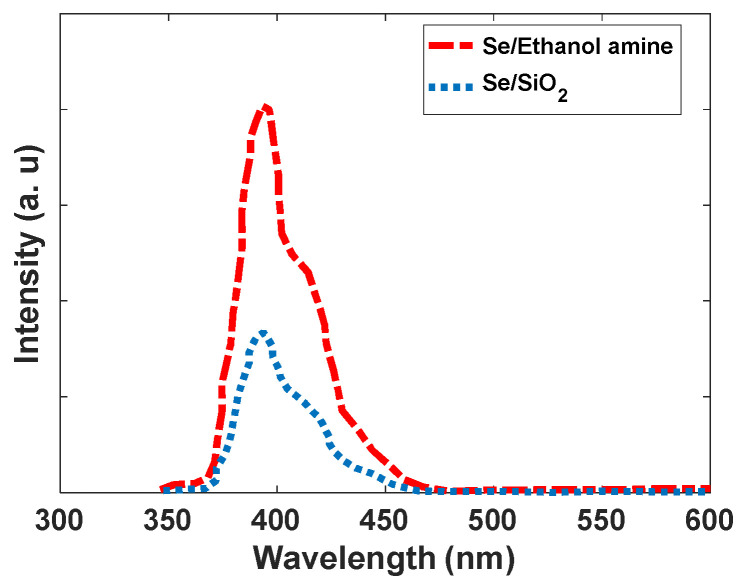
The emission spectra of Se, Se/SiO_2_, and Se/Ethanolamine nanoparticles.

**Figure 15 nanomaterials-12-01423-f015:**
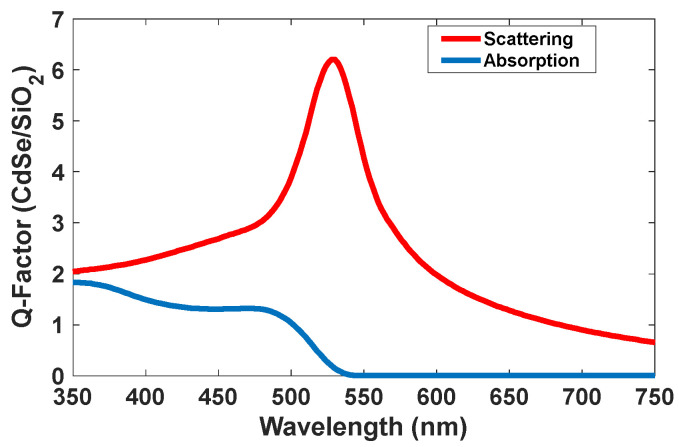
Absorption and scattering spectra of the CdSe/SiO_2_ nanoparticles.

**Figure 16 nanomaterials-12-01423-f016:**
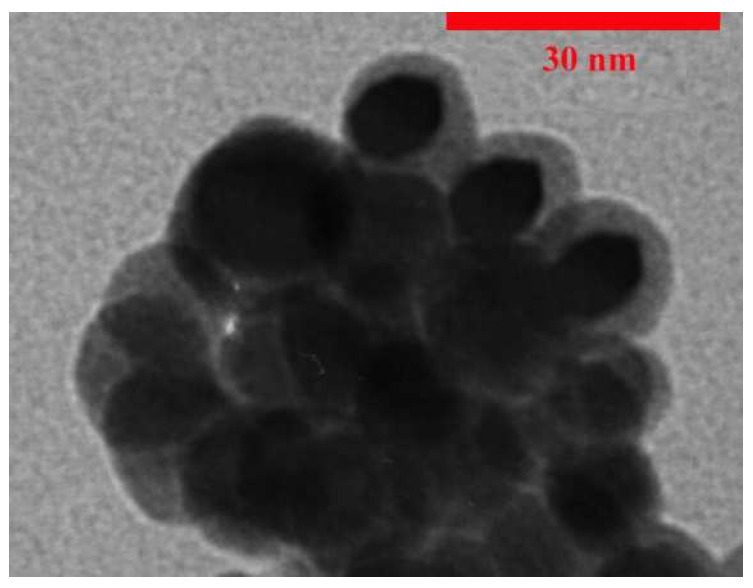
TEM image of the synthesized CdSe/SiO_2_.

**Figure 17 nanomaterials-12-01423-f017:**
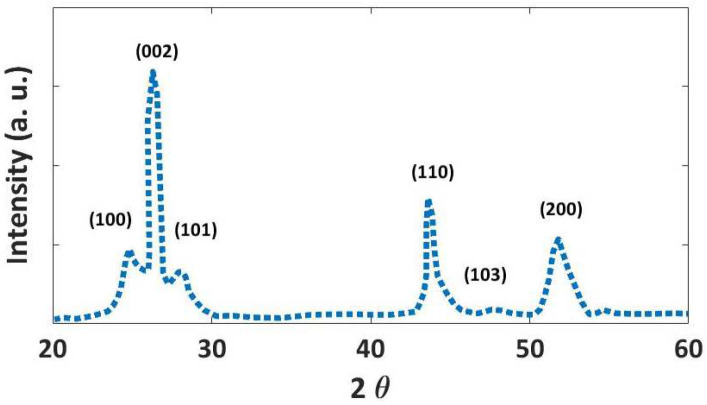
The XRD pattern of the synthesized CdSe/SiO_2_ nanoparticles.

**Figure 18 nanomaterials-12-01423-f018:**
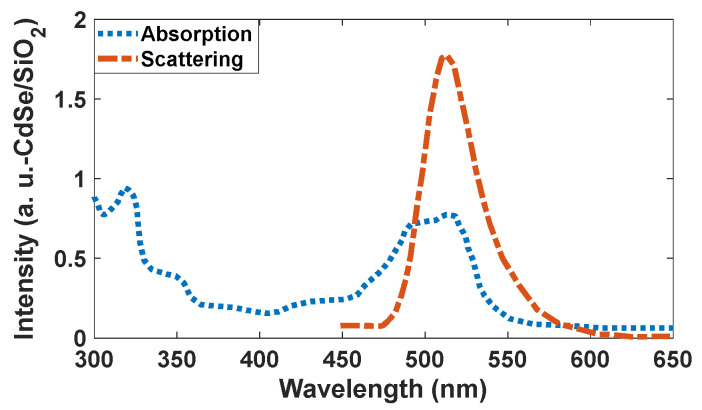
Absorption and scattering spectra of the synthesized CdSe/SiO_2_ nanoparticles.

**Figure 19 nanomaterials-12-01423-f019:**
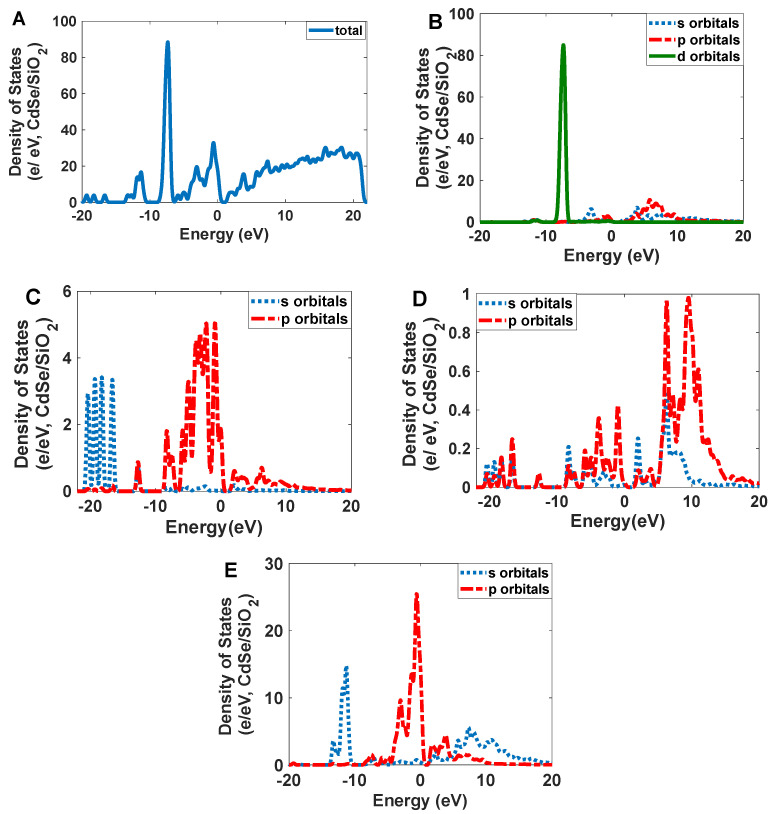
The calculated density of states for CdSe/SiO_2_ nanoparticles: (**A**) total; (**B**) Cd atoms; (**C**) O atoms; (**D**) Si; and (**E**) Se atoms.

**Figure 20 nanomaterials-12-01423-f020:**
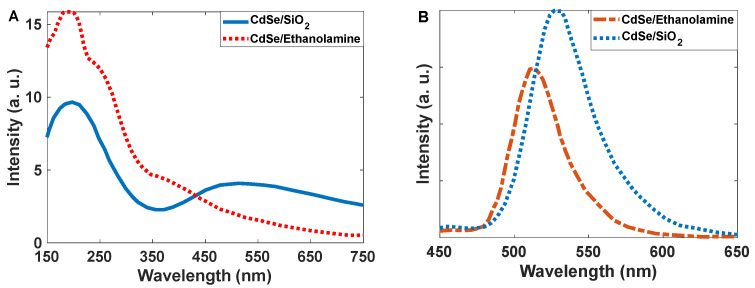
(**A**) Calculated absorption spectra of CdSe/SiO_2_ and CdSe/Ethanolamine nanoparticles; (**B**) scattering spectra of synthesized CdSe/SiO_2_ and CdSe/Ethanolamine nanoparticles.

**Figure 21 nanomaterials-12-01423-f021:**
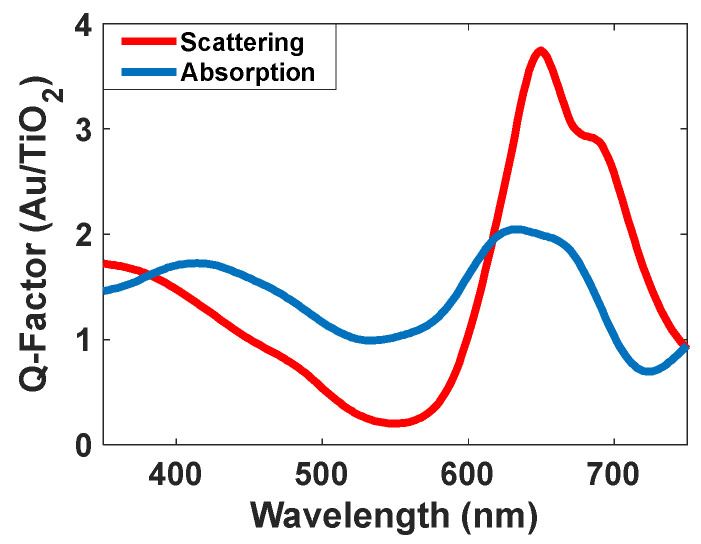
Absorption and scattering spectra of Au/TiO_2_ nanoparticles.

**Figure 22 nanomaterials-12-01423-f022:**
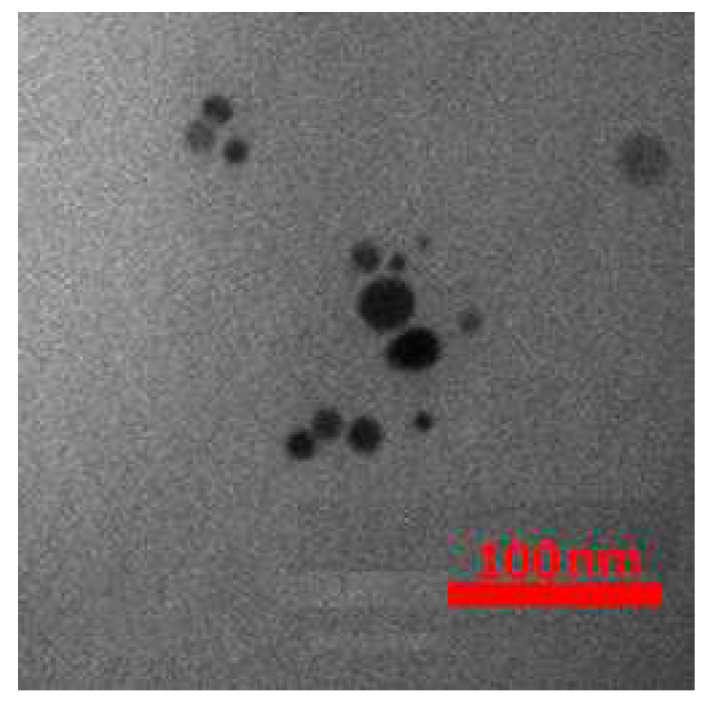
TEM image of the synthesized Au/TiO_2_.

**Figure 23 nanomaterials-12-01423-f023:**
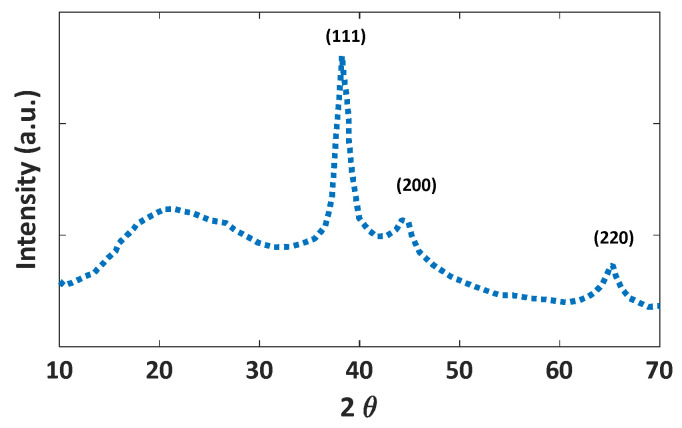
The XRD pattern of the synthesized Au/TiO_2_ nanoparticles.

**Figure 24 nanomaterials-12-01423-f024:**
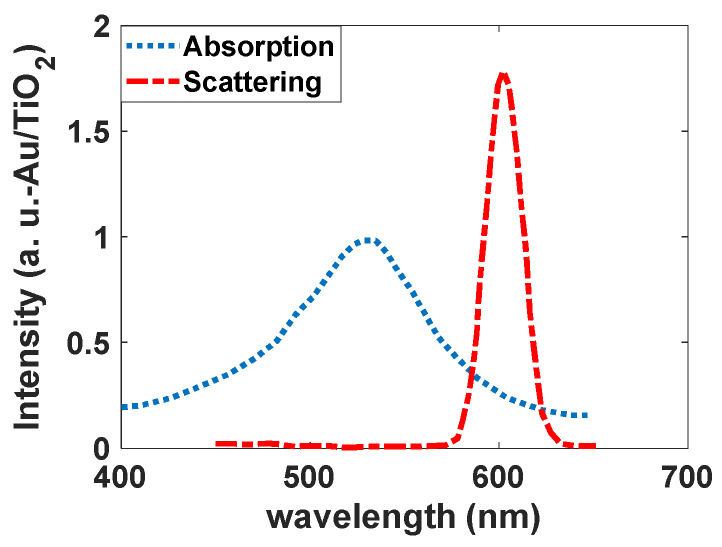
Absorption and scattering spectra of the synthesized Au/TiO_2_ nanoparticles.

**Figure 25 nanomaterials-12-01423-f025:**
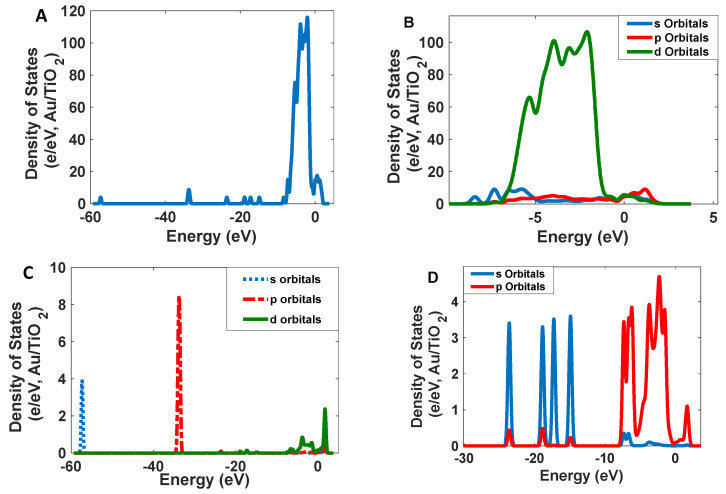
The calculated density of states for Au/TiO_2_ nanoparticles: (**A**) total; (**B**) Au atoms; (**C**) Ti atoms; and (**D**) O atoms.

**Figure 26 nanomaterials-12-01423-f026:**
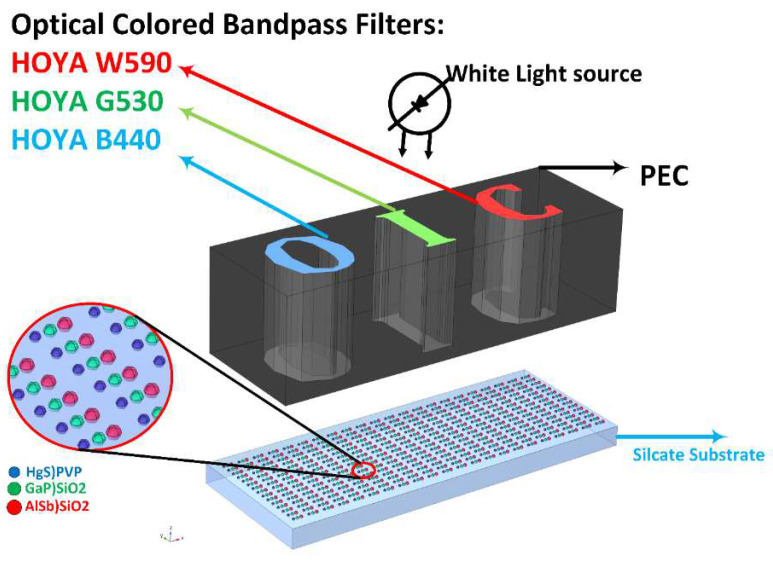
Simulation setup in COMSOL to investigate the light-scattering response of aggregated proposed QDs.

**Figure 27 nanomaterials-12-01423-f027:**
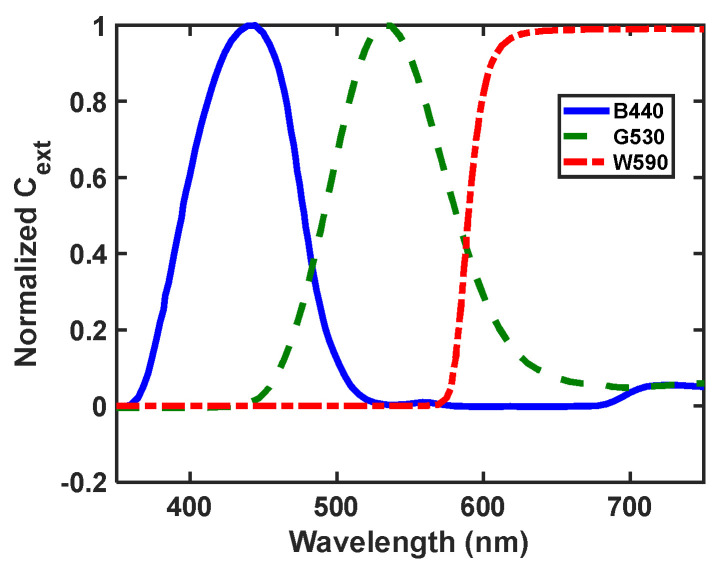
Optical filter profile for R, G, and B, and the optical colored filter demo.

**Figure 28 nanomaterials-12-01423-f028:**
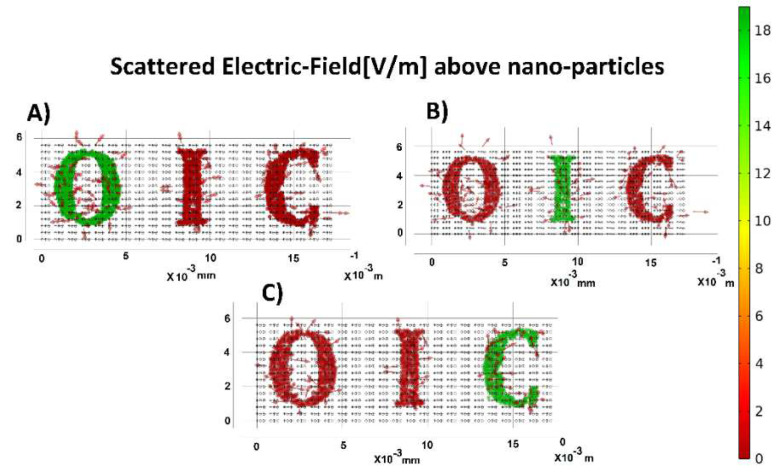
Scattered Field Distribution excited with (**A**) 450, (**B**) 520, (**C**) 630 (nm) incident waves, respectively.

**Figure 29 nanomaterials-12-01423-f029:**
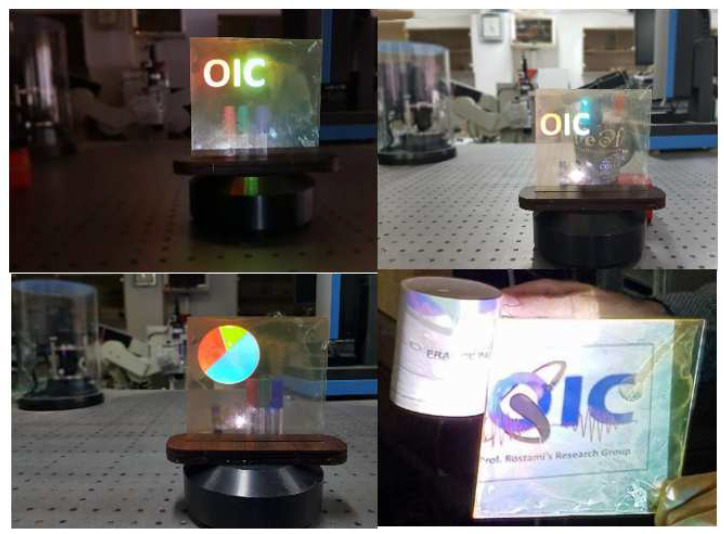
Experimental demonstration of OIC in RGB in dark and daylight conditions.

**Table 1 nanomaterials-12-01423-t001:** Selected nanoparticles for blue, green, and red scattering.

Blue	Green	Red
Ag/SiO_2_ and Ag/TiO_2_	AlAs/SiO_2_ and AlAs/TiO_2_	AlAs/SiO_2_, AlAs/TiO_2_
Si/SiO_2_	CdSe/SiO_2_	AlSb/ SiO_2_, AlSb/ TiO_2_
Se/SiO_2_ and Se/Ethanolamine	Bismuth Silicon Oxide (BSO)/SiO_2_	BiFeO_3_/SiO_2_
BaTiO_3_/SiO_2_	Bismuth ferrite (BiFeO_3_ (BFO))/SiO_2_	BiGaO_3_/TiO_2_
HgS/SiO_2_	InP/TiO_2_	Cd_(1−x)_Eu_x_S/SiO_2_
AlAs/SiO_2_ and AlAs/TiO_2_		Cu/TiO_2_
ZnSe/SiO_2_ and CdS/SiO_2_		Au/SiO_2_

## Data Availability

Not applicable.

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
