# Peer review of "High-Resolution Color Transparent Display Using Superimposed Quantum Dots"

_nanomaterials, 2022, doi:10.3390/nano12091423_

Round 1
Reviewer 1 Report
The manuscript title “High-Resolution Color Transparent Display using Superimposed Quantum
Dots” (Manuscript ID: nanomaterials-1677232) by Mahboubeh Dolatyari , Farid Alidoust, Armin Zarghami, Ali Rostami , Peyman Mirtaheri and Hamit Mirtagioglu, has been revised according to review suggestion and some missing part added.
However, still some improvement must be done mainly related to the quality of some Figures. In particular, all the TEM micrographs have a very poor quality and a scale bar sometimes too prominent. Figure 1, 22 and 24 have a poor resolution, panels in Figure 2 must be aligned and labels should be reported with the same size.
The sentence on line 362-363 -“Gold nanoparticles have interesting optical properties such as surface plasmon resonance (SPR) [39-41], sensing [42-44], photonics [45], and biomedicine which makes fascinating practical applications in various fields [46]” is not correct. Are biomedicine and photonics properties or they are fields of application?
The paper must be also carefully checked .
Author Response
Dear Editor
Enclosed is the revised version of the manuscript and response to reviewers submitted for your consideration.
Bests
Ali Rostami

Reviewer 2 Report
In this manuscript, the authors reported a high-resolution full-color transparent monitor that was designed and fabricated using the synthesized quantum dots. At the first step, Se/SiO2 and BaTiO3/SiO2 nanoparticles are selected as blue, CdSe/SiO2 quantum dot as green, and Au/TiO2 nanoparticles as red QDs for synthesis. In the second step, the surface of nanoparticles is coated with ethanolamine to reveal the role of the surface of the nanoparticles and the electronic trap levels in their structures for increasing the quantum yield and intensity of the light scattering. Furthermore, the authors fabricated high-resolution videos and images displayed on the fabricated monitor. The data are solid and convincing, and the manuscript is well written. Therefore, I would like to recommend its publication in nanomaterials after the authors address the following issues:
- In order to further define the structure and surface morphology of quantum dots, XRD and AFM are needed.
- Some font irregularities appeared in the article. For example, As and s in the eighth line of the abstract were obviously bolded. On page 8, line 7 from the bottom, the first word "with" is also clearly bold.
- Figures are of poor quality (Figure 9). This may be because of the quality of the draft. Please modify these figures in the revised manuscript.
- For transparent display, what is the advantage of working life compared with existing materials?
- Blue, green, and red are mentioned in this paper, which need to be represented by CIE coordinates.
Author Response

(The authors gave the same response as above.)

Reviewer 3 Report
Affiliations need improvement. The number 4 is missing in the description and the reader does not know the abbreviation SP-EPT or OIC
Why some text was highlighted in yellow?
"in the first step" instead of "at the first step"
Abreviation "DFT" should be explained
Figure1 resolution too low
Figure 3 - are the wavelengths for maximum absorption and emission equal to each other?
Figure 4 - there is no distinction between A and B in the figure caption
The dimension marker in Figure 5 is invisible
Figure 14 - The number in dimension marker is too big
PVA abreviation shuld be explained
Conclusion - Authors stated that "The fabricated display was tested for monitoring images and movies." - this is not a conclusion. "Was tested and what was the result os this testing?
Author Response

(The authors gave the same response as above.)

Round 2
Reviewer 3 Report
The authors took note of the comments and the manuscript may be published
This manuscript is a resubmission of an earlier submission. The following is a list of the peer review reports and author responses from that submission.
Round 1
Reviewer 1 Report
This paper is prepared poorly, and I urge the authors to revise carefully about their contents before resubmission to elsewhere. Some of their problems are listed below:
- Only the first letter needed to be capitalized in the whole sentence. This problem has been occurred for many times throughout the manuscript.
- Page 2, line 47: Name of the author “C.W Hsu” is different from that showed in reference 1.
- Page 5, line 198: typo “radiuses”.
- Page 6, subsection 3.1.2: “SiO2” “2” should be placed in subscript.
- Figures 2, 4 and 6: (A) Absorption and (B) Emission spectra should be given to the same chemical species. Their color should be identical so that the interest readers can easily identify their relationship.
- Page 7, subsection 1.7. ZnSe/SiO2 and CdS/SiO2 nanoparticles: I expect the discussion will be focused only on ZnSe/SiO2 and CdS/SiO2 nanoparticles, but this is not the case.
- Similar problem occurred for subsection 2.6. InP/TiO2 nanoparticles.
- Some of the listed quantum dots showed very complicated emission spectrum and I don’t think they can ever be used to generate the monochromatic emission.
- I assumed that the useful nanoparticle should be the core-shell type and with SiO2 or TiO2 shell. However, the authors also spend a lot of their effects in talking about those decorated with et This discussion is totally irrelevant to the current topic.
- The statement showed after page 12 are totally unrelated to the title of the present contribution.
- The paper also has the statement such as “Error! Reference source not found”, it means the paper was poorly prepared.
Reviewer 2 Report
Manuscript titled "High-Resolution Color Transparent Display using Superimposed Quantum Dots" (Muniscript ID: nanomaterials-1628201) by Armin Zarghami, Mahboubeh Dolatyari, Farid Alidoust, Ali Rostami, Peyman Mirtaheri reports a theoretical and experimental work concerning the selection of different semiconductors or metal / dielectric (TiO2 or SiO2) heterostructures for the fabrication of high-resolution color transparent display.
While the Introduction clearly points out the aim of the work and it faces the main issues reported in literature, the results and discussion section is a bit confusing. Even though the authors previously stated that they will focused on a selection of emitting sample, they conversily report the results of different classes of blue, green and red emitting materials. I would suggest the authors to focus on the selected samples and remove all the parts that are not strictly addressed to what they intend to discuss.
On page 6 and throughout the paper, the authors addressed to metal nanoparticles and heterostructures as emitting materials. In view of their property towards application for display technology will it be more rigorous to address to them as materials showing resonance scattering properties rather than fluorescence?
On page 6 line 215 the authors write “The increase in the intensity of emission is higher when the shell is TiO2, in which the peak of the emission spectrum intensity is 7.4; while this is 6.35 in Ag/ SiO2. It should be noted that Ag without a shell has low emission and high absorption intensity. It is because of existing dangling bonds on the surface of nanoparticles that make trap states in the electronic structures of nanomaterials (these types of traps have been described widely in our previous papers)”. Such a sentence has been referred by the authors to Ag@SiO2 nanoparticles, that is based on a metal/dielectric structures. The authors support this conclusion by citing their own paper titled “Trap engineering in solution-processed PbSe quantum dots for high-speed MID- infrared photodetectors, J. Mater. Chem. C, 2019, 7, 5658-5669”, that, conversely describes the behaviour of semiconductor nanoparticles. While for semiconductor such a sentence is reasonable and scientifically correct, I did not find it a valid conclusion to explaing the behaviour of Ag and Ag@SiO2 or Ag@TiO2 nanoparticles.
The description of Figure 12 should be deleted since it does not provide any additional information, but simply reports some already established findings related to luminescence process in nanomaterials.
TEM image of sample CdS@SiO2 has a very poor quality and the shown hetorostructures reveal the formation of a silica matrix, where CdS nanoparticles are probably embedded. The absorption spectra, too, is very different from the abosorption profile characteristic of CdS colloidal nanoparticles, showing an absorption feature at 450-540 nm that could be hardly addressed to CdS nanoparticles. Since the reported energy gap for bulk CdS is 2.4 eV, it would correspond to a exciton emission located at nearly 510 nm. However the authors report an emission spectrum with a maximum located at 550 nm. What kind of phenomena originates such a narrow emission band? If it originates from trap-states, how do the authors can explain the small full width at half maximum of the emission peak?
In the experimental part the equipments used for material characterization (absorption, emission and TEM) are not reported.
According to these criticisms, I would not consider the paper acceptable for publication in the present form. I would recommend major revisions and a thoroughly reorganization of the materials presented in the paper.